# Testing the potential of streamflow data to predict spring migration of an ungulate herds

**Jason S. Alexander**[1]*, **Marissa L. Murr**[2], **Cheryl A. Eddy-Miller**[1]

**1** Wyoming-Montana Water Science Center, U.S. Geological Survey, Water Mission Area, Cheyenne, WY, United States of America, **2** Intern, U.S. Geological Survey-National Association of Geoscience Teachers Cooperative Field Training Program, Cheyenne, WY, Unites States of America

* jalexand@usgs.gov

**Editor:** Stefano Grignolio, University of Ferrara Department of Life Sciences and Biotechnology: Universita degli Studi di Ferrara Dipartimento di Scienze della Vita e Biotecnologie, ITALY

**Data Availability Statement:** All relevant data are within the paper and its Supporting information files. Topographic, hydrologic, and point feature

## Abstract

In mountainous and high latitude regions, migratory animals exploit green waves of emerging vegetation coinciding with rising daily mean temperatures initiating snowmelt across the landscape. Snowmelt also causes rivers and streams draining these regions to swell, a process referred to as to as the 'spring pulse.' Networks of streamgages measuring streamflow in these regions often have long-term and continuous periods of record available in real-time and at the daily time step, and thus produce data with potential to predict temporal migration patterns for species exploiting green waves. We tested the potential of models informed by streamflow data to predict timing of spring migration of mule deer (*Odocoileus hemionus*) herds in a headwater basin of the Colorado River. Models using streamflow data were compared with those informed by traditional temperature-derived measures of the onset of spring. Non-parametric linear-regression techniques were used to test for temporal stationarity in each variable, and logistic-regression models were used to produce probabilities of migration initiation. Our analysis indicates that models using daily streamflow data can perform as well as those using temperature-derived data to predict past-migration patterns, and nearly as well in potential to forecast future migrations. The best performing model was used to generate probabilities of onset of migration for mule deer herds over the 69-year period-of-record from a streamgage. That model indicated spring migration has been trending toward earlier initiations, with modeled median initiations shifting from a Julian day of 123 in the mid 20th century to Julian day 115 over the most recent two decades. The period of 1960 to 1979 had the latest modeled median initiations with Julian day of 128. The analyses demonstrate promise for merging existing hydrologic and biological data collection platforms in these regions to explore timing of past migration patterns and predict migration onsets in real-time.

## Introduction

Exploitation of resource waves has been widely documented as a crucial component in the life history of migratory herbivores [1]. In northern latitudes, so-called 'green-wave' migrations are caused by spatially progressing greening of the landscape along elevation and latitude

data used to create the figures are available from U. S. Geological Survey (https://www.usgs.gov/products/maps/gis-data) and U.S. Department of Agriculture Natural Resources Conservation Service (https://datagateway.nrcs.usda.gov/).

**Funding:** The author(s) received no specific funding for this work.

**Competing interests:** The authors have declared that no competing interests exist.

gradients associated with plant phenological development in the transition from spring to summer [2–5]. The green wave hypothesis is a subset of the broader forage-maturation hypothesis that states ungulates trade off forage quantity for forage quality to maximize energy intake [5, 6]. In mountainous and high latitude regions of North America, the onset of the green wave coincides with rising daily mean temperatures causing snowmelt and subsequent progressive exposure of soil and vegetation along elevation gradients. Snowmelt also causes streams draining these regions to swell giving their annual hydrographs a distinct signature whereby rising flows coincide with greening of the landscape, a phenomenon often referred to as to as the 'spring pulse' [7, 8]. Because streamflow is a point measurement of processes occurring at the scale of a watershed, the spring pulse represents an integrative measure of the green wave across the broader landscape draining to it.

Satellite and temperature-derived measures of the green wave have been used to model timing of migration initiations and destination arrival, characterize plant phenological development along migration routes, and examine various surfing behaviors. The satellite-derived normalized-vegetation difference index (NDVI) and associated derivatives, for example, have been used to model spatial gradients in plant crude protein, making inference on benefits of surfing strategies such as trailing, surfing, or jumping the green wave [3, 9–11]. Bischof et al (2012) [4] used satellite-derived NDVI to observe that male and female red deer (*Cervus elaphus*) moved rapidly from summer to winter grounds, jumping the green wave. Similarly, LaForge et al. (2021) [12] used time series of the instantaneous rate of green up (IRG), an NDVI-derived metric, and observed that female caribou (*Rangifer tarandus*) jumped the greenwave by 1 week, and maximum greenup in summer feeding grounds occurred after calving. Alternatively, Merkle et al. (2016) [2] examined surfing behaviors of several species of ungulates in mountain regions of the western United States and found that a majority of migration paths tracked high IRG areas (i.e. surfed the green wave).

Satellite-derived data such as NDVI, IRG, and snowcover, however, have a limited period-of-record and can have high latency at the daily time step due atmospheric interference (e.g. the Moderate Resolution Imaging Spectroradiometer, MODIS). While typically less limited in period-of-record, temperature-derived measures such as growing degree day require continuous data at the daily time step, and historical temperature data at climate stations have frequent gaps at that time scale. Other temperature-derived metrics, such as the growing degree day 'jerk' [13], require knowledge of the shape of the temperature curve before predictions can be made, limiting their use in forecasting. Spatially extensive daily temperature compilations, such as the Parameter-elevation Regressions on Independent Slopes Model (PRISM; [14]) are substantive improvements over point measures of temperature, and have low-latency, but also have limited periods of record at the daily time step.

Streamgages have operated in the western United States since the late 19[th] century [15] and, like climate stations, can have periods of record with a daily time step spanning many decades and, more rarely, exceeding a century. Today, streamgages operated by the U.S. Geological Survey (USGS) and various state agencies often operate in 'real-time,' transmitting data to the pubic in 15-minute intervals. The extensive periods of record at some streamgages have captured the effects of climate change including earlier onset of the so-called 'spring pulse' in snowmelt-driven streams, declines in shallow winter snowpack causing slower runoff, and declines in the ratio of snow-to-rain, which are projected to increase flood magnitudes [16–20]. Temporal trends in snowmelt and runoff timing have paralleled trends in the timing of plant flowering [7, 8] across the western United States and thus have direct implication for herbivores that exploit green waves. There is evidence, for example, that earlier snowmelt and

higher spring temperatures negatively affect plant phenological development and diversity, and reduce offspring production of migrating herbivores [21, 22].

Data on animal land-migration patterns across North America are now becoming more widely available [23], and these datasets present opportunities to more closely examine linkages between hydroclimate-driven landscape processes and animal migrations. Here we test potential of models informed by minimally transformed, publicly available data to predict timing of spring migration of mule deer. We use measures of greenwave processes that have a long period-of-record (>30 years) and low-latency at the daily time step with a goal of examining temporal trends in migration initiation and demonstrating potential for prediction of migration at the daily time step. These purposes are beneficial to practical applications such as traffic safety, land-use planning, herd management, migration fencing management, and predicting climate change effects on migrations, but also work towards a broader goal of the USGS of building a multi-disciplinary integrated data and knowledge platform [24]. We hypothesized that the timing of the spring pulse measured at a streamgage would closely correspond to the timing of onset of spring migration of mule deer in a cold-desert river basin and would demonstrate potential for linkages between data being measured at streamgages and animal migrations. We compare models that use temperature-derived measures of plant phenological development with those using the timing of spring snowmelt to predict spring migration of groups within two mule-deer herds in the Colorado River Basin.

## Materials and methods

Traditional measures of plant phenological development were extracted from gridded daily mean temperature data, and the timing of spring pulse initiation was extracted from daily mean streamflow data. Non-parametric linear-regression techniques were used to test for temporal stationarity in these variables. Logistic regression models were used to examine the potential to predict initiation of spring migration of individuals and groups of female mule deer in the same broader river basin.

### Study area

The study area is the migratory range of two populations of female mule deer (*Odocoileus hemionus*) within the Baggs herd of Wyoming [23]. In keeping with previous studies [25–27], these populations are respectively referred to as the northern and southern Atlantic Rim herds (*herein* NAR and SAR herds, respectively). Female mule deer in these herds have been studied extensively and have been shown to be surfers of the green wave, have very few residents, and low plasticity in their migration routes [2, 26, 28]. Although female deer in these herds have been the primary focus of study, past research has noted that spatial segregation between male and female deer occurs at a fine scale, and that migration patterns among males are likely similar to females [29]. Nonetheless, the analysis, results, and conclusions presented in this paper apply only to female mule deer because no male mule deer were tracked.

The migratory range of the NAR and SAR herds lie almost entirely within the Little Snake River Basin in south-central Wyoming and northwest Colorado (Fig 1). The Little Snake River drains approximately 9,700 km$^2$ of montane and sagebrush-steppe landscapes of the Colorado River headwaters [30, 31]. Elevation in the basin ranges from 1,710 to 3,350 meters and annual precipitation ranges from 190 to 435 mm, with large proportions falling as snow in the highest elevation regions. The montane regions include sub-alpine forests, alpine meadows, and steep, rocky alpine slopes. Sagebrush steppe regions are high-elevation cold deserts characterized by rolling topography with dry valleys and intermittent streams.

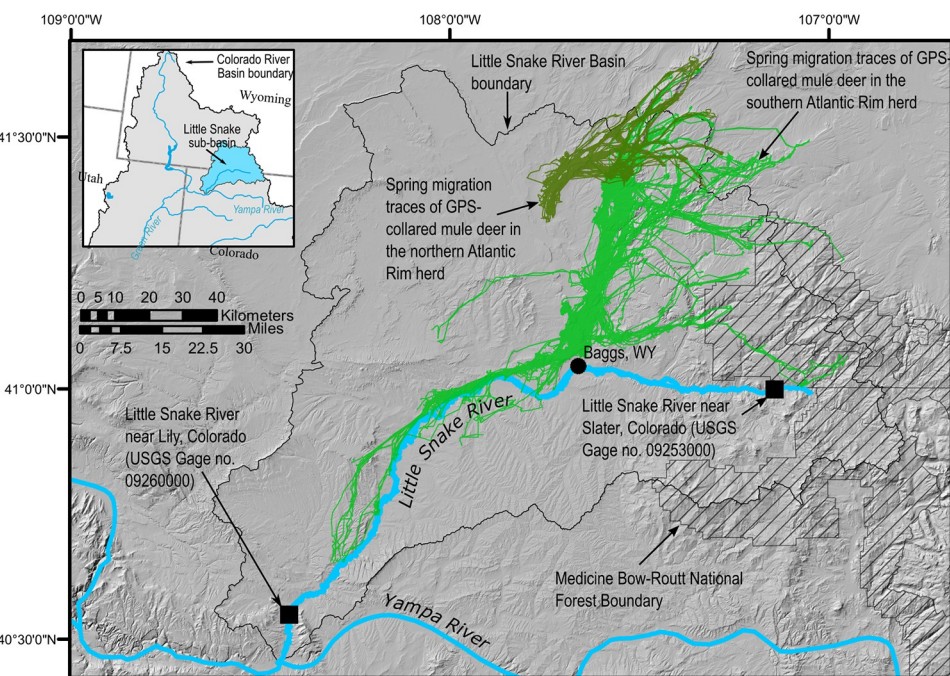

**Fig 1. Map of the Little Snake River Basin in south-central Wyoming and northwest Colorado including the locations of the migration routes of the southern and northern Atlantic Rim mule deer herds and streamgages used for the analysis.** Topographic, hydrologic, and point feature data used to create the map are all freely available from U.S. Geological Survey (https://www.usgs.gov/products/maps/gis-data) and U.S. Department of Agriculture Natural Resources Conservation Service (https://datagateway.nrcs.usda.gov/). Migration paths are from [23].

## Data

Three primary datasets were used as the basis for analysis: (1) daily mean surface temperature; (2) daily mean streamflow; and (3) spring migration data from radio-collared female mule deer in the two Atlantic Rim herds. Temperature data were extracted for the entire Little Snake River Basin area. Streamflow data were extracted from the USGS's National Water Information System (NWIS; [32]) for two streamgages in the basin: (1) the Little Snake River near Lily, Colorado (USGS streamgage no. 09260000), and (2) the Little Snake River near Slater, Colorado (USGS streamgage no. 09253000; Fig 1). The streamgage near Slater is located at the edge of undeveloped national forest and represents a generally natural snowmelt hydrograph, whereas the streamgage near Lily is located near the downstream end of the basin and is affected by various streamflow diversions. Both streamgages have relatively long daily periods of record with limited data gaps.

**Mean basin temperature and derivatives.** Daily mean basin temperature ($\langle T_i \rangle$) was calculated using the PRISM *AN81d* dataset ([14]). These data are available free of charge in gridded format at 4-kilometer resolution, with a daily period-of-record extending to the present from January 1981. Gridded temperature data were subset to the basin area, and $\langle T_i \rangle$ was calculated as the simple arithmetic mean of the subset for each day *i* (Fig 2).

Growing degree day (*GDD*) for each year *j* was calculated for the basin using the method of Van Wijk and others (2012; [13]), whereby the *GDD* of a certain Julian day *i* is the accumulation of growing degree units (*GDU*) leading up to that day beginning on January 1:

$$GDD_{ji} = \sum_{i=1}^{k} GDU_{ji} \tag{1}$$

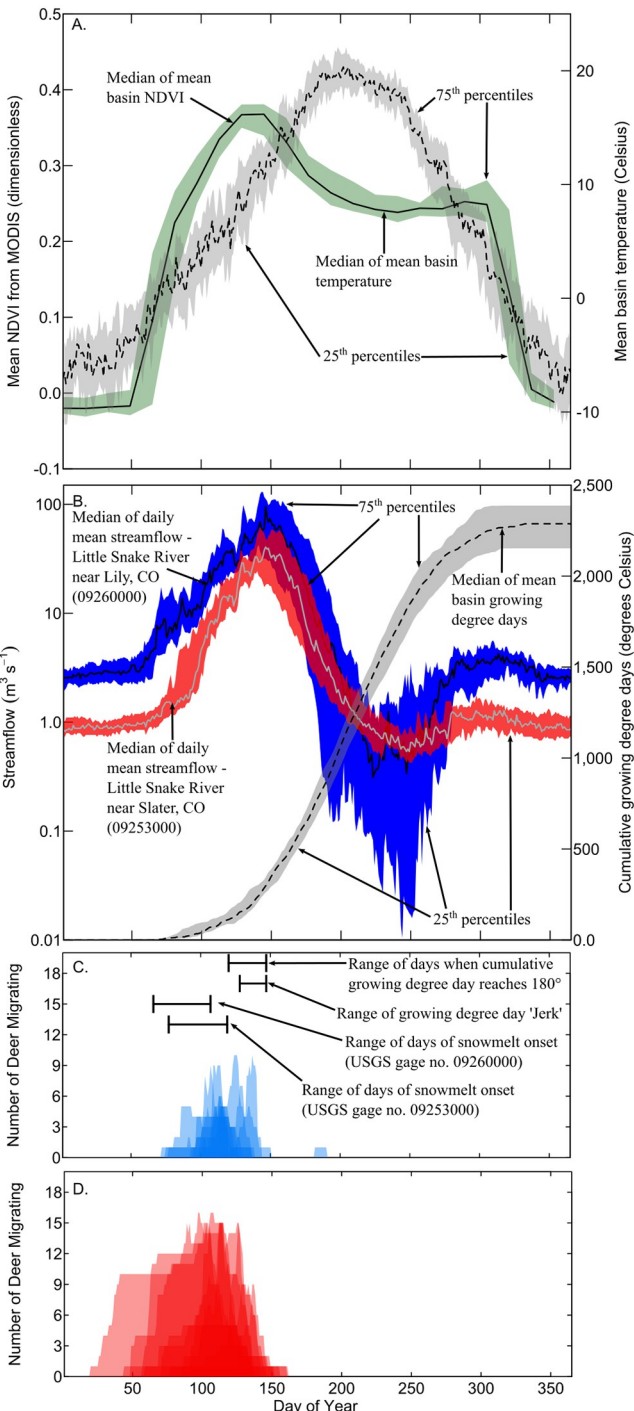

**Fig 2. Time series of medians and inter-quartile ranges of (A) Mean basin Normalized Difference Vegetation Index (NDVI) and mean basin daily temperature, and (B) daily mean streamflow and mean basin cumulative growing degree day; (C) Number of radio-collared female mule deer in spring migration for the northern Atlantic Rim herd; and (D) Number of radio-collared female mule deer in spring migration for the southern Atlantic Rim herd.** Mean basin NDVI, daily temperature, and cumulative growing degree day are for the entire Little Snake River Basin. Streamflow data are from two U.S. Geological Survey streamgages. Growing degree days and temperature are from the PRISM dataset. NDVI data are from the National Aeronautics and Space Administration's Moderate Resolution Imaging Spectroradiometer (MODIS) satellite platform and are shown here because of the widespread use of NDVI as an indicator of greenwave processes. NDVI data were not included in the analysis. The data in (A) and (B) are for the 2002 to 2018 period of record. The data in (C) and (D) are for years 2005–2006, 2008–2010, 2015–2018, with darker colors indicating overlap in timing between years.

and

$$GDU_{ji} = \langle T_{ji} \rangle - T_{base} \tag{2}$$

The base temperature was calculated as $T_{base} = -0.25 * latitude + 13$. For the purposes of trend analysis, the first day of spring in each year $j$ was taken as the day that $GDD_{ji}$ exceeded 180°C.

The maximum rate of spring warmup was calculated for each year $j$ as the so-called growing-degree day 'jerk' ($GDDjerk_j$), a physics term used to describe the rate of change in acceleration [13]. The method described by Van Wijk and others (2012; [13]) was used to calculate $GDDjerk_j$, whereby the maximum of the third derivative of a sigmoid function fit to a $GDD$ curve for each year $j$ was taken as the day of $GDDjerk_j$. Sigmoid functions were fit to annual time series of $GDD_{ji}$ using self-starting logistic models in language $R$ [33]. Parameter estimates from these models were used to extract the third derivative of the sigmoid function and identify the $GDDjerk_j$ for each year $j$. Again, for the purposes of modeling spring migration initiation in mule deer, $GDDjerk_j$ was converted to a continuous variable of daily time step by taking the difference with Julian days:

$$JERK_{ji} = J_{ji} - GDDjerk_j \tag{3}$$

where $J_{ji}$ is Julian day $i$ in year $j$.

**Timing of snowmelt pulse initiation.** The method of Cayan et al. (2000; [7]) was used to identify the Julian day of the onset of the annual snowmelt pulse ($D_{sm_j}$) at each streamgage. The method uses the cumulative departure of daily mean streamflow from the annual mean streamflow, and thus requires the calendar-year annual mean streamflow be known *a priori*. For the purposes of modeling, this variable was transformed to a daily time step by taking the difference between the Julian days $i$ in year $j$ and $D_{sm_j}$:

$$D_{sm_{ji}} = J_{ji} - D_{sm_j} \tag{4}$$

To explore the potential of $D_{sm_{ji}}$ to predict spring migration initiation at the daily time step, we calculated a second explanatory variable, $Dp_{sm_{ji}}$, which uses the previous calendar year's annual mean streamflow as the reference for cumulative departure. This lagged version of $D_{sm_{ji}}$ was only calculated for the Little Snake River near Slater, Colorado (USGS streamgage no. 09253000; Fig 1) because $Dp_{sm_{ji}}$ was strongly correlated (Pearson's R = 0.70) to $D_{sm_{ji}}$ at that streamgage (Fig 3).

**Onset of spring migration of mule deer.** The binary response variable in our logistic regression modeling was onset of spring migration of groups of mule deer $MH_{ji}$ within a herd. These data originated from radio-collar tracking of female mule deer in the NAR and SAR herds over nine years (2005–2006; 2008–2010, 2015–2018). Migration initiation days were identified using the method of net-squared displacement [34], and full descriptions of the migration data, including initiation dates, can be found in Kaufman et.al (2020; [23]).

We arbitrarily defined the day that groups of deer (proxy for herds) initiated migration as the first day of each year $j$ the maximum number of collared deer in each herd were observed migrating simultaneously. That is, if the maximum number of deer migrating simultaneously in the NAR herd in year $j$ was 7, then $MH_{ji}$ was set to 1 on the first Julian day at least 7 deer in the NAR herd were simultaneously in spring migration that year. We applied these rules to the first 151 days of the year whereby all days prior to migration initiation were set to a value of 0, and all days after initiation up to Julian day 150 were set to 1. This 151-day period, although

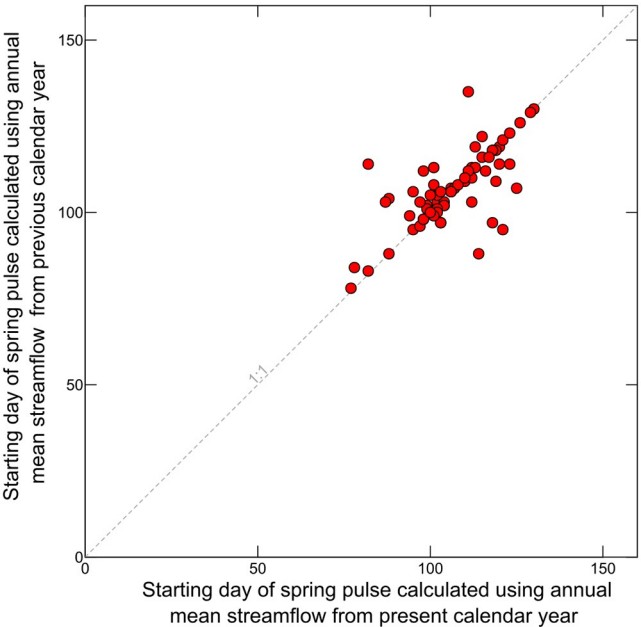

**Fig 3. Scatterplot showing comparison of calculated day of onset of the spring pulse using the annual mean streamflow from the present calendar year with that calculated using the annual mean streamflow from the previous year.** All data are for the Little Snake River near Slater, Colorado (USGS streamgage no. 09253000) for 1944 to 2018 period-of-record. The day of the spring pulse was calculated using the method of Cayan et al. (2000; [7]).

arbitrary, was chosen because it is the end of May in common years, and encompassed the latest calculated $GDD_{180}$ and $GDD_{jerk}$ (147), latest day of spring pulse initiation (119), and latest day of initiation of group migration within mule deer herds (137).

## Modeling

The non-parametric Kendall-Theil regression method [35, 36] was used to test for temporal trends in $GDD_{180_j}$ and $GDD_{jerk_j}$, and the timing of snowmelt ($D_{sm_j}$) in each year *j* spanning their respective periods of record. The data were checked for autocorrelation before testing for temporal trends, and no significant autocorrelation was detected in any of the series.

Logistic regression models were used to assess the relative potential of explanatory variables to predict onset of spring migration of individuals ($MI_i$) and groups ($MH_i$). Logistic regression was used because we were most interested in representing migration initiation as a probability. This interest is rooted in a recognition that the response variable, initiation of spring migration in female mule deer, is informed by data from tens of animals, at most, collectively representing individuals numbering in the thousands. Our data model assumed $MI_i$ and $MH_i$ were Bernoulli-distributed with probability of initiation of spring migration denoted by $\theta_i$, and logit of $\theta_i$ was modeled as a linear function of our explanatory variables:

$$Mx_i|\theta_i \sim Bern(\theta_i) \tag{5}$$

$$logit(\theta_i) = \beta_o + \ldots + \beta_{k_i} \tag{6}$$

where $logit(\theta_i)$ is the *logit* link function $log\left(\frac{\theta_i}{1-\theta_i}\right)$, and $\beta_k$ are model coefficients associated with our explanatory variables, including a dummy variable to identify mule deer herd (NAR or

SAR). Pearson's R was used to examine the potential effect of multicollinearity on model explanatory variables. Pearson's R values greater than 0.20 between explanatory variable were considered to have high potential for effects of multicollinearity when included in the same model.

The logistic models were implemented using a Bayesian framework in language *R* [33] and the Gibbs sampling package 'Rjags' [37]. To minimize the effect of priors on parameter estimates, diffuse priors were assumed for model parameters (i.e. $\beta_k \sim N\left[\mu_k = 0, \frac{1}{\sigma} = 0.0001\right]$). Gibbs sampling used 1,000 burn-in steps with 3 Markov-chains of 20,000 steps each to update prior distributions and generate posterior credible distributions for $\beta_k$. Model convergence was assessed using plots of Markov-chain traces, autocorrelation, and Rubin-Gelman scale reduction factors. Model performance was assessed using posterior deviance information criteria [38], recall, and precision of posterior predictive distributions.

## Results

### Trends in temperature and snowmelt indicators of spring

Non-parametric regression models show tendencies toward earlier spring across all measures of spring onset, but the slope coefficient was only significant ($p < 0.05$) for the spring pulse at the Little Snake River near Slater, Colorado (USGS streamgage no. 09253000; 'Slater streamgage'). At that streamgage, the trend indicates the snowmelt pulse is 1.7 days earlier with each decade (Fig 4), equating to a mean shift of nearly 13 days over the 1943 to 2018 period of record. The Little Snake River near Lily, Colorado (USGS streamgage no. 09260000; 'Lily streamgage') had a stream record dating back to 1921, but the daily record did not span the entire year until 1932, and that was the first year used to calculate timing of a snowmelt pulse there. The Lily streamgage had a non-significant slope of -0.6 days per decade over the 1932 to 2018 period of record. A non-significant slope of about 1 day per decade was estimated for $GDDjerk_j$, while $GDD_{180_j}$ had a slope of -0.6 per decade. The slope parameter for the Slater streamgage was non-significant when the data was modeled over the period-of-record overlapping with $GDD_{180_j}$ and $GDDjerk_j$ (1981 to 2018).

### Characteristics of spring migration of mule deer in the Atlantic Rim herds

Spring migration initiation dates and displacement distances for individual collared female mule deer in the SAR and NAR herds varied by individuals and from year to year (Table 1). The number of females tracked in any given year varied, but was at least 7 except in 2010, when only 1 individual in the NAR herd was tracked. Collared individuals in the SAR herd universally initiated spring migration earlier than individuals in the NAR herd (Fig 5). Individual collared female mule deer initiated spring migration as early as January 20[th] in the SAR herd, and as early as March 13[th] for the NAR herd. Median difference between migration initiation date across all years was 36 days earlier for individuals in the SAR herd relative to the NAR herd. Displacement calculations across all years indicate individuals in the SAR herd migrated a median of 20 km farther than individuals in the NAR herd.

Although spring migration initiation dates of individuals in the NAR and SAR herds showed substantial differences, very little difference was found for groups of mule deer within each herd (Fig 5). Our measure of herd migration initiation ($MH_i = 1$) used the first day the maximum number of individuals in each herd were in migration simultaneously as the initiation date for the broader respective herds. In all but two years for one herd, more than 60% of the collared deer were migrating simultaneously, indicating our measure of group migration

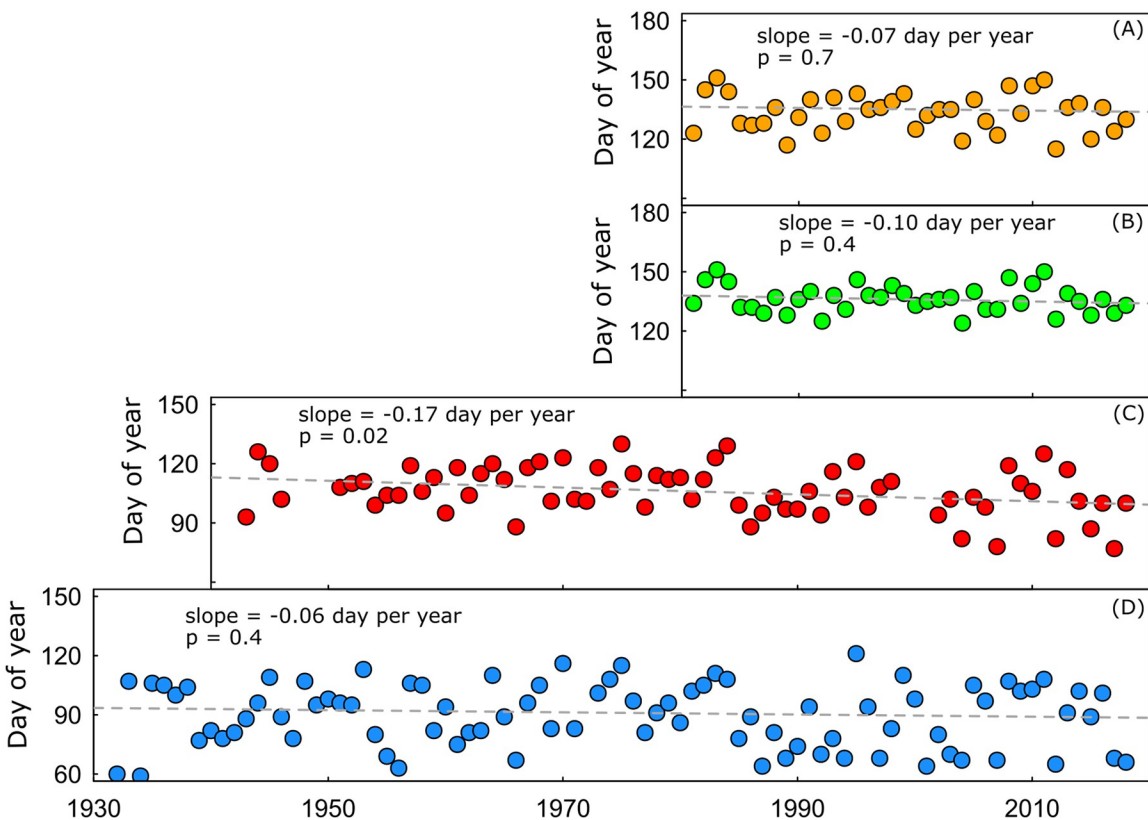

**Fig 4. Time series showing temperature-dependent measures of plant phenology and annual snowmelt over time for the Little Snake River of north-central Colorado and south-central Wyoming.** (A) Julian day when cumulative growing degree day exceeds 180˚C; (B) Julian day when cumulative growing degree days are increasing most rapidly, commonly referred to as the 'growing degree day jerk;' (C) Julian day of onset of spring snowmelt pulse for the Little Snake River near Slater Colorado (USGS streamgage no. 09253000); (D) Julian day of onset of spring snowmelt pulse for the Little Snake River near Lily Colorado (USGS streamgage no. 09260000). Slopes shown were estimated using the non-parametric Kendall-Theil regression method.

initiation date was a reasonable indicator of herd behavior. Across all years, the median difference in migration initiation day between groups within the NAR and SAR herds was 1 day.

## Logistic models of initiation of spring migration of mule deer groups

Logistic modeling results demonstrate all measures of spring initiation were significant predictors of mule deer migration (Table 2). All explanatory variables were also strongly correlated, with minimum Pearson's correlation coefficient of 0.78 across all combinations. The models including growing degree day (*GDD*) and the day of snowmelt initiation for the Slater streamgage ($D_{sm}^{slater}$) had the best overall performance for groups of female mule deer (Table 2). The model using $D_{sm}^{slater}$ as the singular explanatory variable for onset of migration in female mule deer groups ($MH_i$) had the lowest deviance information criteria (DIC), and the highest values of recall and precision of all single-variable models. Although DIC decreased slightly for multi-variate models including $D_{sm}^{slater}$, this came at the expense of recall and precision, indicating the models were not substantively improved with the addition of the temperature-derived measures.

Models using *GDD*, *JERK*, $\langle T_i \rangle$, $D_{sm}^{lily}$ (day of snowmelt pulse initiation for the Lily streamgage), and $Dp_{sm}^{slater}$ (day of previous year's snowmelt pulse initiation for the Slater streamgage)

**Table 1. Data describing radio-collared female mule deer of the Atlantic Rim herds of northwest Colorado and south-central Wyoming.**

| Herd | Year | n | Indiv. mig. start date[1] | Herd mig. start date[2] | Max n mig. simult.[3] | Median displacement distance (km)[4] |
|------|------|---|----------------------------|--------------------------|------------------------|--------------------------------------|
| NAR | 2005 | 7 | 4/9 | 4/23 | 6 | 32 |
| NAR | 2006 | 8 | 4/3 | 4/21 | 5 | 28 |
| NAR | 2008 | 12 | 4/30 | 5/16 | 10 | 26 |
| NAR | 2009 | 8 | 3/18 | 5/8 | 7 | 23 |
| NAR | 2010 | 1 | 5/5 | 5/5 | 1 | 16 |
| NAR | 2015 | 11 | 3/16 | 3/27 | 5 | 34 |
| NAR | 2016 | 14 | 4/4 | 5/3 | 10 | 34 |
| NAR | 2017 | 16 | 3/13 | 4/17 | 9 | 21 |
| NAR | 2018 | 15 | 4/11 | 4/19 | 9 | 24 |
| SAR | 2005 | 21 | 4/1 | 4/16 | 16 | 37 |
| SAR | 2006 | 20 | 3/27 | 4/22 | 15 | 36 |
| SAR | 2008 | 9 | 3/17 | 5/15 | 9 | 30 |
| SAR | 2009 | 19 | 2/28 | 5/6 | 15 | 35 |
| SAR | 2010 | 11 | 3/12 | 5/8 | 9 | 56 |
| SAR | 2015 | 24 | 1/20 | 4/7 | 16 | 56 |
| SAR | 2016 | 20 | 2/26 | 4/22 | 15 | 68 |
| SAR | 2017 | 18 | 2/18 | 4/11 | 15 | 63 |
| SAR | 2018 | 19 | 2/13 | 4/22 | 15 | 46 |

Explanation: mig.—migration; NAR—northern Atlantic Rim herd; SAR—southern Atlantic Rim herd;

[1] earliest migration day of radio-collared female mule deer;

[2] earliest migration day when maximum number of radio-collared female mule deer were migrating simultaneously;

[3] maximum number of radio-collared female mule deer migrating simultaneously;

[4] straightline distance between starting and ending locations of spring migration.

as singular explanatory variables had slightly lower performance in predicting migration onset of groups of female mule deer, with $\langle T_i \rangle$ having the highest DIC, and lowest recall and precision (Table 2). The performance of models including $Dp_{sm}^{slater}$ increased slightly, but not substantively, with the addition of the correlated variables $\langle T_i \rangle$ and *GDD*. Although the addition of $\langle T_i \rangle$ would likely have increased the performance of models with *GDD* and *JERK* as singular explanatory variables, we did not attempt to model this because their base level of performance was already lower than $D_{sm}^{slater}$ and these explanatory are derived from and strongly correlated to $\langle T_i \rangle$. As expected, parameter estimates on the dummy variable identifying herd were not significant, indicating none of the models could distinguish a signal in spring migration timing between groups within herds.

## The snowmelt hydrograph and timing of spring migration of mule deer

Model 5 (Table 2) was used to generate probabilities of initiation of spring migration for mule deer groups in the NAR and SAR herds for calendar years 2002 to 2018. This period of record was chosen because it is the most recent continuous period of record of daily streamflow at the Slater streamgage overlapping with the period of mule deer migration observations (Fig 6). Model 5 was used because, despite model 7 having slightly lower DIC, model 5 had equivalent recall and precision, was more parsimonious, and avoided issues of multicollinearity. The 95% probability regions generated using model 5 show reasonable agreement with observed migration initiation dates for individuals and herds of deer, whereby probabilities in these regions

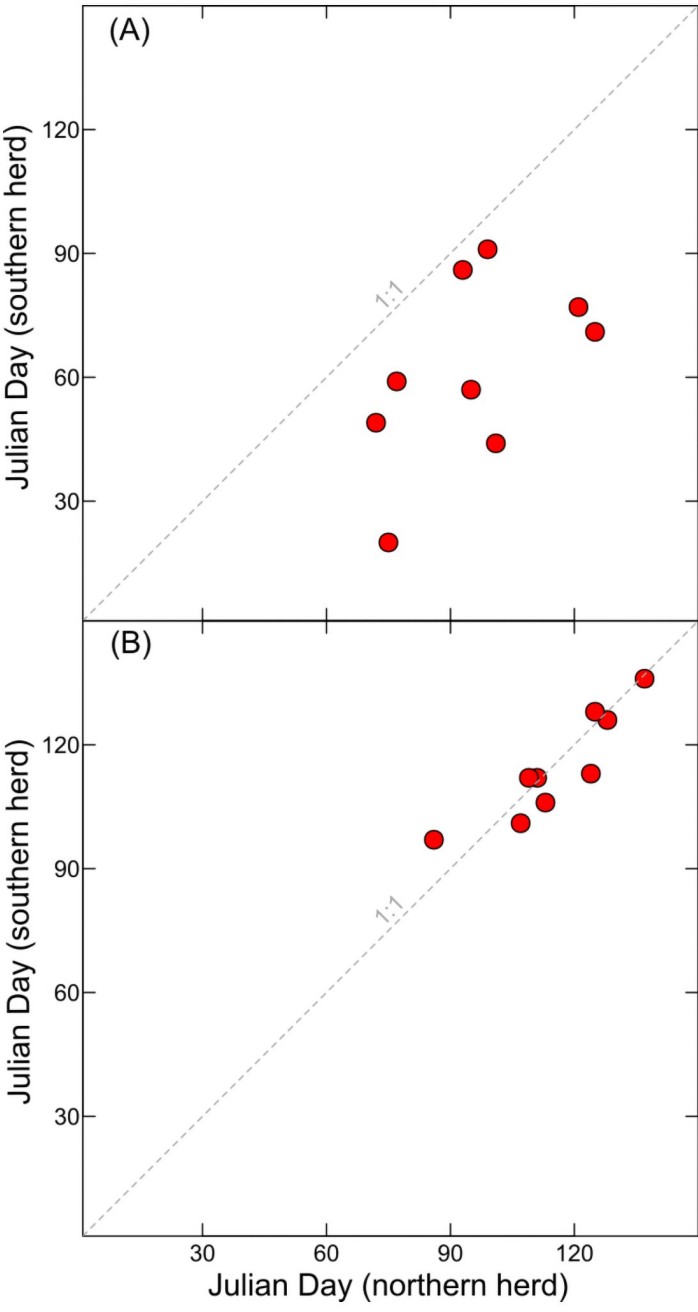

**Fig 5. Scatterplot showing comparison of the Julian day of onset of spring migration for (A) individuals, and (B) groups of radio-collared female mule deer within the northern (NAR) and southern (SAR) Atlantic Rim herds.** All data are for the years 2005 to 2006, 2008 to 2010, and 2015 to 2018. The days shown in (A) are the earliest migration day of a radio-collared individual female mule deer within each herd. The days shown in (B) are the earliest day when the maximum number of radio-collared female mule deer were migrating simultaneously.

are almost universally greater than 50% on the days of observed migration initiation for collared female deer.

To directly compare our model performance with observations, and to make estimates over the historical period of record, the first day in which median posterior predicted probabilities

**Table 2. Results of logistic regression models—Parameter and error estimates.**

| Model | Response | $\langle T \rangle$ | GDD | JERK | $D_{sm}^{lily}$ | $D_{sm}^{slater}$ | $Dp_{sm}^{slater}$ | Herd | DIC | Recall | Prec. |
|---|---|---|---|---|---|---|---|---|---|---|---|
| 7 | $MG_i$ | **0.13** | – | – | – | **0.25** | – | -0.27 | 472 | 0.92 | 0.93 |
| 5 | $MG_i$ | – | – | – | – | **0.25** | – | -0.25 | 486 | 0.92 | 0.93 |
| 2 | $MG_i$ | – | **0.07** | – | – | – | – | -0.23 | 538 | 0.90 | 0.92 |
| 9 | $MG_i$ | **0.11** | **0.02** | – | – | **0.19** | – | -0.27 | 465 | 0.90 | 0.91 |
| 8 | $MG_i$ | **0.13** | – | – | – | – | **0.15** | -0.17 | 736 | 0.89 | 0.91 |
| 10 | $MG_i$ | **0.07** | **0.05** | – | – | – | **0.05** | -0.25 | 517 | 0.88 | 0.91 |
| 3 | $MG_i$ | – | – | **0.20** | – | – | – | -0.20 | 608 | 0.90 | 0.90 |
| 6 | $MG_i$ | – | – | – | – | – | **0.16** | -0.16 | 761 | 0.88 | 0.89 |
| 4 | $MG_i$ | – | – | – | **0.13** | – | – | -0.14 | 876 | 0.85 | 0.86 |
| 1 | $MG_i$ | **0.39** | – | – | – | – | – | -0.07 | 1616 | 0.66 | 0.73 |

Explanation: Model numbers correspond to the order the models were run. Models are ordered from top to bottom by priority of highest precision and highest recall, and lowest DIC. Parameter values in boldface indicate Bayesian posterior 95% credible interval does not contain 0; $MG_i$—Group migration initiated\not initiated; $\langle T \rangle$—mean basin temperature; GDD—growing degree day; JERK—growing degree day 'jerk'; $D_{sm}^{lily}$—day of snowmelt pulse initiation for Little Snake River near Lily, Colorado (USGS streamgage no. 09260000); $D_{sm}^{slater}$—day of snowmelt pulse initiation for Little Snake River near Slater, Colorado (USGS streamgage no. 09253000); $Dp_{sm}^{slater}$—day of previous year's snowmelt pulse initiation for Little Snake River near Slater, Colorado; Herd—dummy variable identifying northern or southern Atlantic Rim herds; DIC—deviance information criteria; Recall—fraction of actual positives identified as positive, calculated as ratio of true positives to sum of true positives and false negative; Prec.—precision: fraction of actual positives included in predicted positives calculated as ratio of true positives to sum of true positives and false positives.

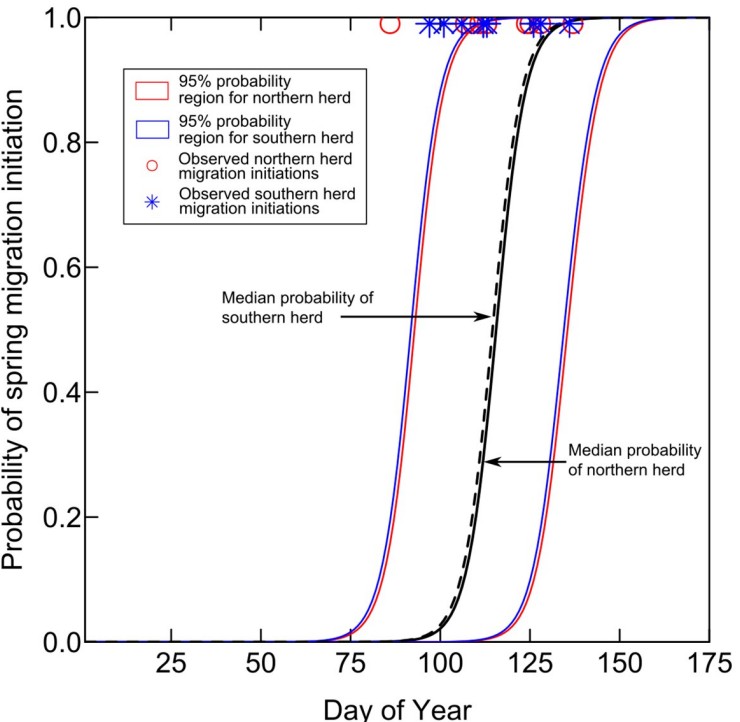

**Fig 6. Probabilities of spring migration initiation for mule deer in the Atlantic Rim herds of south-central Wyoming and northwest Colorado.** The probability regions were generated by logistic regression using the timing of annual onset of the snowmelt pulse in the Little Snake River, near Slater, Colorado (USGS streamgage no. 09253000) as the primary explanatory variable. The period of record used to generate the probabilities was 2002 to 2018, which is the most recent continuous period of record. Water years 2000 and 2001 had incomplete records. The observations of initiation of spring migration for herds of mule deer are from years 2005–2006, 2008–2010, 2015–2018.

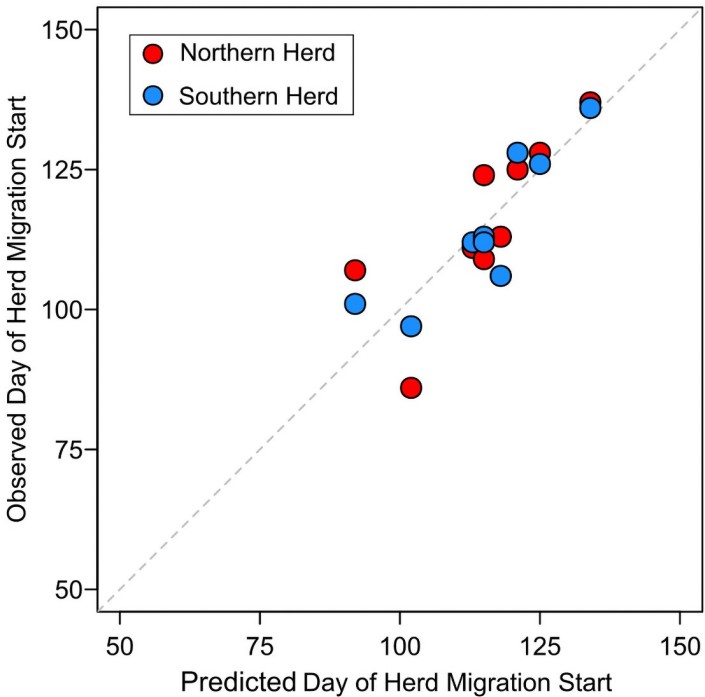

**Fig 7. Predicted versus observed day of initiation of migration for mule deer in Atlantic Rim herds of south-central Wyoming and northwest Colorado.** Years included are 2005–2006, 2008–2010, 2015–2018.

exceeded 50% was used as the predicted day of spring migration initiation for groups of mule deer in the NAR and SAR herds. For the years with observations of mule deer migrations, there is reasonable agreement between predicted and observed migration initiation days (Fig 7). The slope parameter on predicted versus observed data in Fig 7 is 0.95, indicating our model has a tendency to predict migration initiation slightly later in the year than observed. Root-mean square error (RMSE) between simulated and observed data is 7.4 days.

Using the 1943 to 2018 period-of-record available for the Slater streamgage on the Little Snake River, model 5 was also used to examine how the timing of spring migration initiation of the Atlantic Rim herds may have changed over time. That analysis indicated that the timing of mule deer spring migration initiation has become earlier with time (Fig 8). The median predicted Julian day for spring migration initiations of mule deer groups was 123 (May 2–3) for the period 1943 to 1959, 128 (May 7–8) for the period 1960 to 1979, 118 (April 27–28) for the period 1980 to 1999, and 115 (April 24 to 25) for the period 2000 to 2018. Inter-quartile ranges for the same periods were 9, 16, 15, and 19 days, respectively.

## Discussion

The models included in the analysis were intentionally simple and were constructed mainly to compare performance of models using streamflow records with those using more traditional measures of spring to predict past and future green-wave migration timing of female mule deer. Given the limited available data, the models demonstrate accurate reproduction of observed migration behaviors among groups within the NAR and SAR herds. The model using timing of the spring pulse measured at the Slater streamgage ($D_{sm}^{slater}$) had the best overall performance, but it is unclear if streamgages in other basins with green-wave migrations

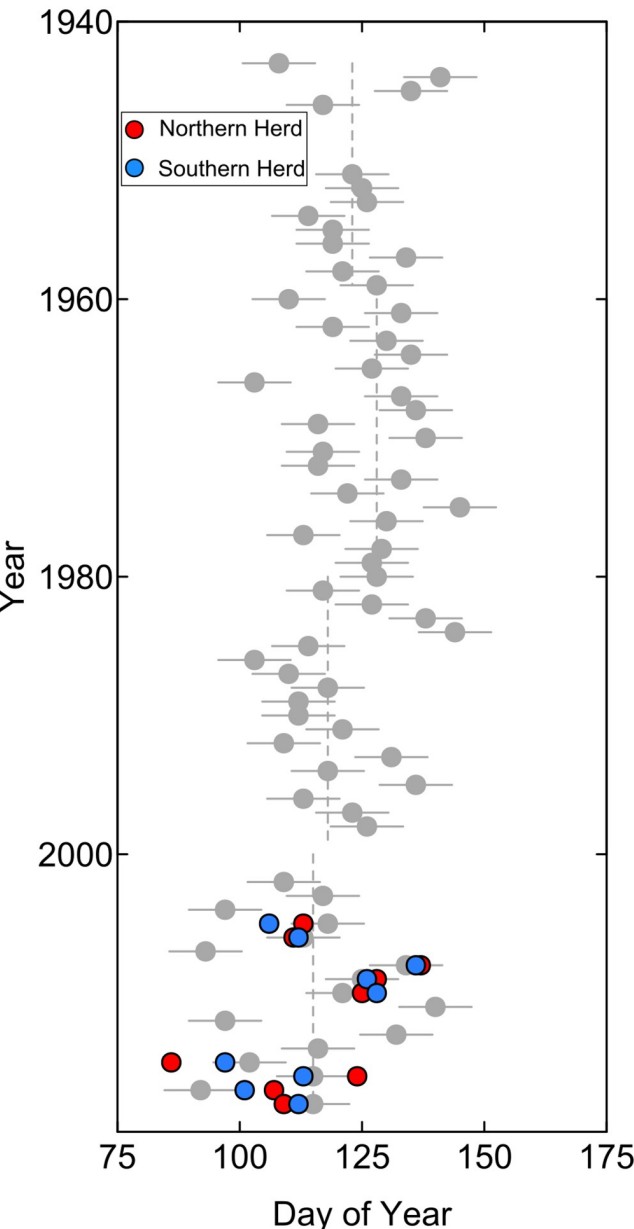

**Fig 8. Predicted historical and modern day spring migration initiation for mule deer in the northern and southern Atlantic Rim herds.** The gray dots show predicted day with error bars showing standard errors. Red and blue circles are observed migration initiation dates for the northern and southern herds, respectively. Dashed gray lines are medians for respective time periods.

would have the same performance. Models that included snowmelt timing of the Lily streamgage ($D_{sm}^{lily}$), a location that represents snowmelt for a broader region of the basin, had weaker performance than models using traditional measures of spring onset (*GDD* and *JERK*). It is possible that this weaker performance was due to $D_{sm}^{lily}$ being more affected by streamflow diversions, or that the lower elevation portions of the Little Snake River Basin have more complex snowmelt dynamics than $D_{sm}^{slater}$. It is also possible that the runoff dynamics at $D_{sm}^{slater}$ more accurately represented the progression of the green wave along the migration routes and\or near

the summer feeding grounds of the Atlantic Rim mule deer herds. Additional testing of relations between migration initiation in other herds with networks of streamgages in basins along or near migration routes and feeding grounds would be beneficial to understanding limits of the types of hydrology-based models presented herein.

Although our model parameter estimates are significant and accurately reproduce migration behaviors in the NAR and SAR herds, our sample size was relatively small (n = 17), and there are inherent assumptions in our models that add complexity to interpretation of our results. The modeled prediction of a median shift in migration initiation of 8 days earlier than the mid-20th century for groups of female mule deer is just beyond the calculated error of the best model (RMSE = 7.4 days). That model relied only on the timing of snowmelt initiation, and thus was strongly influenced by the underlying, but significant, temporal trends of that data. Modeling temporal trends in the timing of female mule deer migration over the past 85 years required assumptions that these herds have been responding to the same environmental cues over that period, that no major shifts in migratory behaviors have occurred (i.e. switch between resident and migratory), and that locations and distances between each herd's summer and winter feeding grounds has not changed. Given that land uses, land and herd management, and vegetation across this landscape has undoubtedly changed over that time period, this assumption is potentially naive. Research by Sawyer et al. (2019) [28] has shown that mule deer in the mountainous western United States, have little plasticity in migratory routes, indicating some of our assumptions are reasonable for these species, but may not hold for species with higher migratory route plasticity. Recent modeling by Barnhardt et al. (2020) [39] has indicated that snowmelt timing and rate are modulated by plant-available water and available energy, which adds complexity to the assumption that the response of spring plant phenology to snowmelt has remained constant over the modeled historical period.

The data in our analysis also demonstrate parity with observations of mule deer as so-called green-wave 'surfers', whereby migration is concurrent with the leading edge of vegetation exposure on the landscape [2, 40]. Animals consistently exhibiting surfing behavior might be expected to delay migration until enough snowmelt has occurred to expose and allow some growth of vegetation along their migratory paths [3]. Indeed, simple linear least-squares regression between our calculated day of spring migration onset of NAR and SAR herds and the day of snowmelt onset produced a linear equation with slope of 0.95 days per day and intercept of 20 days. This relation indicates a consistent delay of nearly three weeks between the onset of snowmelt and onset of migration, with a slight decrease in the lag for later snowmelt onset. With these considerations in mind, our analysis further demonstrates the strong phenological cue that snowmelt provides animals in northern and high-altitude climates. This strong coupling is also further evidence of phenological plasticity to changing environmental cues [4, 12, 41]. Such phenological plasticity has also been demonstrated in species native to temperate climates, and may act as a buffer to inevitable environmental changes [42].

The extensive and continuous periods of record at some streamgages invites potential for investigating temporal changes in migration initiation that may assist in understanding population dynamics of migrating ungulates that exploit the green wave. The advantage of using streamflow data for the 0predictions demonstrated here is the combination of convenience, latency, and period-of-record. The publicly available gridded-temperature data used in our analysis (PRISM) are convenient and straightforward to extract through the PRISM database, but the period-of-record for the data is limited to 1981 to the present. Additional temporal richness could have been obtained through analysis of various climate stations within the basin. There are several National Weather Service climate stations that record temperature in various locations in the Little Snake River Basin. However, only one station has a period-of-record pre-dating the 1970s, and that station's period-of-record ends in 1978 (Dixon,

Wyoming). This alone would have required developing a a separate, hierarchical statistical relationship with other station to extend the period of record.

Our modeling efforts also show promise for prediction in real-time. Models 6 and 8 (Table 2), which use the date of the previous year's Julian day of onset of snowmelt pulse, indicate only slightly decreased performance relative to models using $D_{sm_{ji}}$, *GDD*, and *JERK*. When combined with $\langle T_i \rangle$, which is also available on a daily time step, model performance is slightly improved. With additional migration data spanning a larger spatial domain, higher-level machine-learning models could potentially further improve performance by leveraging additional information from all hydroclimatic variables available at the daily time step to improve forecasting performance. With the recent availability of migration patterns of numerous ungulates in northern latitudes, there is substantial potential for informing higher-level models that incorporate streamflow and climate data. Such models would provide a basis for forecasting potential changes in ungulate migration timing and patterns under a changing climate.

## Supporting information

**S1 Data.**
(TXT)

## Acknowledgments

Thanks to Hall Sawyer and Western EcoSystem Techology Inc. for providing an earlier compilation of the Atlantic Rim mule deer migration data. Any use of trade, firm, or product names is for descriptive purposes only and does not imply endorsement by the U.S. Government.

## Author Contributions

**Conceptualization:** Jason S. Alexander, Marissa L. Murr, Cheryl A. Eddy-Miller.

**Formal analysis:** Jason S. Alexander, Marissa L. Murr.

**Funding acquisition:** Cheryl A. Eddy-Miller.

**Methodology:** Marissa L. Murr.

**Project administration:** Cheryl A. Eddy-Miller.

**Supervision:** Cheryl A. Eddy-Miller.

**Validation:** Jason S. Alexander.

**Writing – original draft:** Jason S. Alexander, Marissa L. Murr, Cheryl A. Eddy-Miller.

**Writing – review & editing:** Jason S. Alexander, Cheryl A. Eddy-Miller.

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
