## [Decision Letter · Decision Letter 0]

17 Feb 2021

PONE-D-20-40787

Testing the potential of streamflow data to predict spring migration of an ungulate

PLOS ONE

Dear Dr. Alexander,

Thank you for submitting your manuscript to PLOS ONE. After careful consideration, we feel that it has merit but does not fully meet PLOS ONE’s publication criteria as it currently stands. Therefore, we invite you to submit a revised version of the manuscript that addresses the points raised during the review process.

We look forward to receiving your revised manuscript.

Kind regards,

Zaher Mundher Yaseen

Academic Editor

PLOS ONE

Journal Requirements:

2. Please ensure that you refer to Figure 8 in your text as, if accepted, production will need this reference to link the reader to the figure.

3.We note that Figure(s) 1 in your submission contain map images which may be copyrighted. All PLOS content is published under the Creative Commons Attribution License (CC BY 4.0), which means that the manuscript, images, and Supporting Information files will be freely available online, and any third party is permitted to access, download, copy, distribute, and use these materials in any way, even commercially, with proper attribution. For these reasons, we cannot publish previously copyrighted maps or satellite images created using proprietary data, such as Google software (Google Maps, Street View, and Earth). For more information, see our copyright guidelines: http://journals.plos.org/plosone/s/licenses-and-copyright.

a) You may seek permission from the original copyright holder of Figure(s) 1 to publish the content specifically under the CC BY 4.0 license. 

Additional Editor Comments:

Authors shall revise their manuscript as per the reported comments.

Reviewers' comments:

Reviewer's Responses to Questions

**Comments to the Author**

1. Is the manuscript technically sound, and do the data support the conclusions?

Reviewer #1: Partly

Reviewer #2: Yes

2. Has the statistical analysis been performed appropriately and rigorously? 

Reviewer #1: No

Reviewer #2: Yes

3. Have the authors made all data underlying the findings in their manuscript fully available?

Reviewer #1: Yes

Reviewer #2: Yes

4. Is the manuscript presented in an intelligible fashion and written in standard English?

Reviewer #1: Yes

Reviewer #2: Yes

5. Review Comments to the Author

Reviewer #1: The authors provided an approach to predict the timing of spring migration of mule deer herds and individuals based on the Colorado River's streamflow data. The subject of the paper is interesting, and it has the potential to publish in the PONE. Since the method's application is the only contribution of the current study, in this stage of review, I only focus on the method section. If the paper has a chance for further consideration based on the editor's decision, I will provide other comments after checking the other sections. However, the authors should address several concerns in this stage of review. So, I recommend the paper should be revised as major revision.

- The authors should provide a flow chart to demonstrate the method proposed in the current study step by step.

- The authors used the Kendall-Theil method to assess the trend of the variables employed in the current study. What is the advantage of that method? Is there any other alternative to conduct trend analysis. What is the advantage. I believe the author should use recently developed methods.

- The authors used a simple linear regression model for prediction modeling. I believe the relationship between the target and predictive variables is not linear. So, I suggest examining the AI models for prediction.

- Please update the results base on the mentioned comments as well.

Reviewer #2: In the present paper, the authors applied the Logistic-regression model for modelling the timing of spring migration of mule deer herds. The goal fixed by the authors was clear and objectively justified: predicting the onset of spring migration for herds of mule deer in south-central Wyoming. One of the most significant results of the present investigation is that the streamflows data was more suitable as predictor compared to the temperature data for predicting past and future migrations. The paper is well written, scientifically sound and the originality is evident as the research gap is clearly presented and argued. Clear explanation of methods, results and discussion, and the topic of the manuscript are very important. As such, this manuscript definitely deserves to be published in the PLOS ONE Journal. I have only one fewer suggestions to the authors:

1. Page 5/14 Lines: For the purposes of modeling we defined the day of herd migration initiation as the first day the maximum number of deer in each herd were in spring migration. I think this sentence needs to be reformulated.

2. Figures 3 and 7 (Scatterplot) provides the R2 and the regression equation.

3. Figure 2 should be improved it is for bad quality and unclear.

6. PLOS authors have the option to publish the peer review history of their article (what does this mean?). If published, this will include your full peer review and any attached files.

Reviewer #1: No

Reviewer #2: **Yes: **Pr. Salim Heddam

---

## [Author Response · Author response to Decision Letter 0]

3 Mar 2021

Our responses to reviewers and the editorial staff are contained in the document titled 'Response to Reviewers.pdf'.

---

## [Decision Letter · Decision Letter 1]

18 Mar 2021

PONE-D-20-40787R1

Testing the potential of streamflow data to predict spring migration of an ungulate

PLOS ONE

Dear Dr. Alexander,

Thank you for submitting your manuscript to PLOS ONE. After careful consideration, we have decided that your manuscript does not meet our criteria for publication and must therefore be rejected.

I am sorry that we cannot be more positive on this occasion, but hope that you appreciate the reasons for this decision.

Yours sincerely,

Zaher Mundher Yaseen

Academic Editor

PLOS ONE

Additional Editor Comments (if provided):

The authors are advised to address the reviewers comments and resubmit the manuscript.

Reviewers' comments:

Reviewer's Responses to Questions

**Comments to the Author**

1. If the authors have adequately addressed your comments raised in a previous round of review and you feel that this manuscript is now acceptable for publication, you may indicate that here to bypass the “Comments to the Author” section, enter your conflict of interest statement in the “Confidential to Editor” section, and submit your "Accept" recommendation.

Reviewer #1: (No Response)

Reviewer #2: All comments have been addressed

2. Is the manuscript technically sound, and do the data support the conclusions?

Reviewer #1: Partly

Reviewer #2: Yes

3. Has the statistical analysis been performed appropriately and rigorously? 

Reviewer #1: No

Reviewer #2: Yes

4. Have the authors made all data underlying the findings in their manuscript fully available?

Reviewer #1: No

Reviewer #2: Yes

5. Is the manuscript presented in an intelligible fashion and written in standard English?

Reviewer #1: Yes

Reviewer #2: Yes

6. Review Comments to the Author

Reviewer #1: Unfortunately, the authors did not addressed all of my comments in the previous stage. The paper in the current is not supported with a robust methodology. Actually, the Kendall method and simple linear regression modeling are not adequate for a scientific paper in 2021. So, I suggest to reject the paper in the current form and encourage the authors to resubmit the paper as new submission when they will use the newly developed methods/models.

Reviewer #2: The authors have correctly addressed all comments and the paper is ready for publication. The par is scientifically sound and well organized.

7. PLOS authors have the option to publish the peer review history of their article (what does this mean?). If published, this will include your full peer review and any attached files.

Reviewer #1: No

Reviewer #2: **Yes: **Pr. Salim Heddam

- - - - -

---

## [Author Response · Author response to Decision Letter 1]

26 Mar 2021

Please see our attached file 'Response to Reviewers.pdf' for our responses.

---

## [Decision Letter · Decision Letter 2]

5 Jul 2021

PONE-D-20-40787R2

Testing the potential of streamflow data to predict spring migration of an ungulate

PLOS ONE

Dear Dr. Alexander,

Thank you for submitting your manuscript to PLOS ONE. After careful consideration, we feel that it has merit but does not fully meet PLOS ONE’s publication criteria as it currently stands. Therefore, we invite you to submit a revised version of the manuscript that addresses the points raised during the review process.

We look forward to receiving your revised manuscript.

Kind regards,

Stefano Grignolio, Ph.D

Academic Editor

PLOS ONE

Journal Requirements:

Additional Editor Comments (if provided):

Reviewers' comments:

Reviewer's Responses to Questions

**Comments to the Author**

1. If the authors have adequately addressed your comments raised in a previous round of review and you feel that this manuscript is now acceptable for publication, you may indicate that here to bypass the “Comments to the Author” section, enter your conflict of interest statement in the “Confidential to Editor” section, and submit your "Accept" recommendation.

Reviewer #3: (No Response)

Reviewer #4: (No Response)

2. Is the manuscript technically sound, and do the data support the conclusions?

Reviewer #3: Partly

Reviewer #4: Yes

3. Has the statistical analysis been performed appropriately and rigorously? 

Reviewer #3: I Don't Know

Reviewer #4: Yes

4. Have the authors made all data underlying the findings in their manuscript fully available?

Reviewer #3: Yes

Reviewer #4: Yes

5. Is the manuscript presented in an intelligible fashion and written in standard English?

Reviewer #3: Yes

Reviewer #4: Yes

6. Review Comments to the Author

Reviewer #3: The authors provided a new method to predict female mule deer migration onset from streamflow data which are available in real time and for long term time series, undoubtedly opening interesting perspectives for wildlife management. The idea is very original and the logical flow easy to follow. Nevertheless, in my opinion some points should be clarified before publication. The reported anticipating trend of migration onset should be statistically justified, but my impression is that the potential to predict future (rather than past) migrations is the most important and useful finding of the manuscript. Below you can find more specific comments on major and minor issues to be addressed.

Major issues:

- Introduction and Discussion chapters miss information regarding cues of ungulate migrations. The reported past and future predictions on mule deer migration onset is justified only if environmental cues (temperature, snowmelt, vegetation growth…all variables that authors cleverly summarized within the streamflow) are the only (or main) cue triggering migration onset. Nevertheless, phenology of northern latitudes ungulate species typically (mostly) relies on photoperiodism, which is independent form environmental changes. This is the reason for global change-driven mismatches between migration and optimal foraging conditions provided within the green wave (e.g., Stenseth and Mysterud 2002; Mysterud 2013) and the consequential loss of fitness in migrating ungulates (mentioned by the authors in L46). Notwithstanding, there are evidence of high environmental plasticity of the migration timing, for example in caribou which mostly trigger migration with snowmelt (Laforge et al. 2021). The results of this manuscript potentially provide a similar evidence, but this issue should be introduced and discussed within its broader chronobiological context.

- The onset of spring migration by individuals (MI) is biased by the variability in the number of individuals monitored for each year and herd. With the described method, larger samples will inevitably have a higher likelihood to pick smaller minimum values (in this case, earlier dates). MIji is in other words not comparable among years and herds (and indeed seems anticipated in SAR, the herd with more monitored individuals). As individual and herd migration dates are inconsistent (i.e., when comparing SAR and NAR) and predictions have been rightly based only on herd migration (MH) I would suggest removing any mention of MI from the manuscript.

- Predictions of past migrations (i.e., occurred between 1943 and 2005) assume that the importance of environmental cues (in respect to photoperiodism, see previous comment) in triggering mule deer migrations has remained unchanged since 1943 despite the ongoing climate change. This can be a reasonable assumption but, in my opinion, should be explicitly communicated to readers, as we cannot exclude that mule deer population became more and more capable of using environmental cues (and not photoperiodism) to trigger migration.

- The anticipating trend of migration onset from 1943 to 2018 should be statistically supported rather then only described in terms of medians of arbitrary periods of 20 years. As a matter of example, just suppose that this study was conducted in 1979: authors would have concluded that along 36 years, from 1943 to 1979, the migration onset had been delayed of 5 days (rather than anticipated because of climate change). Looking at Figure 8, my impression is that the variability of predicted migration onset dates among years, together with their uncertainty, do not support the conclusion of a tendency to anticipate migrations over years (but my impression could be proven wrong with a proper statistical test).

Minor issues:

- Introduction miss information about the known migration patterns of the monitored species. In particular:

• Do mule deer behave as a green wave “surfer” or “jumper”? This information is needed to understand if they can simply follow the green wave or rather must anticipate it with reliable cues.

• Does the whole population migrate every year, or a proportion of individuals are stationary? Is this eventual proportion constant among years or rather it varies depending on environmental conditions? This is important to understand the biological meaning of the estimated MH.

• Do females and males migrate together? Or, in other words, are female migration patterns a good proxy of the whole population migration? As only female mule deer has been monitored, this should be stated and has major management implications. If there are no reasons to consider female migration as a reliable proxy of the whole populations, the authors should specify throughout the text that their results only refer to female mule deer.

- Just for clarity, how was the database organized for the logistic regression models? If I understood correctly, it was organized with 151 records per year (one for each considered Julian date) assuming 0 from 1st January until the MH date of the reference year and 1 from MH date onward. Please consider rephrase the corresponding Methods section to clarify.

- Figure 6: y axes are named (slightly) different in A and B. In addition, it is unclear why predictions were made for the period 2003-2018, while observations counted only 9 years and started in 2005. Please clarify.

Literature cited:

Laforge, M. P., Bonar, M., & Vander Wal, E. (2021). Tracking snowmelt to jump the green wave: phenological drivers of migration in a northern ungulate. Ecology, 102(3), e03268.

Mysterud, A. (2013). Ungulate migration, plant phenology, and large carnivores: The times they are a‐changin'. Ecology, 94(6), 1257-1261.

Stenseth, N. C., Mysterud, A., Ottersen, G., Hurrell, J. W., Chan, K. S., & Lima, M. (2002). Ecological effects of climate fluctuations. Science, 297(5585), 1292-1296.

Reviewer #4: In this ms, Alexander and colleagues test the applicability of hydrological measurements informed by streamflow data to assess their potential in predicting the timing of migration of mule deer in North America. I received a revises version of the ms, for further revision. I liked the ms, which combines two field of research (movement ecology and hydrology) in a promising way. I think it deserves publication in PlosONE. It is well written, it clearly states the problem, it has a strong logical structure, and it presents interesting results. As movement ecologist, I don’t feel competent to judge the technical details of the methodological process concerning the hydrological data. Instead, I have very minor suggestions about the ‘ecological side’ of the work, that I hope can further improve the quality of the ms.

1) First line of the abstract: I would remove ‘of North America’: green wave surfing is a movement tactic that applies to ungulates worldwide

2) LL 18-20: when you speak about NDVI applied to measure surfing capacity by ungulates, I would recommend to cite the following work:

Aikens, E. O., Kauffman, M. J., Merkle, J. A., Dwinnell, S. P., Fralick, G. L., & Monteith, K. L. (2017). The greenscape shapes surfing of resource waves in a large migratory herbivore. Ecology Letters, 20(6), 741-750.

3) LL 69-71: “Logistic regression models were used to examine the potential to predict initiation of spring migration of individuals and groups of mule deer in the same broader river basin”. Please replace ‘groups’ by ‘herds’

4) LL 164-165: “Logistic regression models were used to assess the relative potential of explanatory variables to predict onset of spring migration of individuals (MIi) and groups (MHi)”. I think that it would be appropriate to move here in this section the issue that you raise about correlation between covariates (instead of presenting it in the Results, LL 229-230): such way, the reader comes to the results sections with a bit clearer idea about what models have been fitted and compared.

5) Results of the models (lines 226 and following): I would suggest separating the part of results where describe the outcome of models for group migration vs individual migration. It seems that you start with a sort of ‘mix’ (“The models including growing degree day (GDD) and the day of snowmelt initiation for the Slater streamgage (Dslater sm ) had the best overall performance for groups and individuals of mule deer”); then, you move to describe the group migration results, but this is not immediate to catch, and only later it becomes clear (i.e. when you move to the individuals results description). It is just a recommendation to help clarity of presentation.

6) Caption to Figure 5: “Herd - dummy variable identifying northern or southern Atlantic Rim herds” � Although you say it in the main text, I would specify also here what is the reference category

7) Table 2: I suggest to present the list of models based on their ranking for precision, it is easier to follow

8) LL 263-265: “As stated previously, we were most interested in representing spring migration initiation of mule deer as a probability because we believe it is the most reasonable representation of data which are, at best, under-informed.” � I think this sentence is useless and it does not add any information, I would remove it.

9) L301 and following: I appreciate the tentative to discuss the different results obtained for the two streamgates. Do you think this could also due to a simple issue of distance between the streamgate and the tracking animals? If this is the case, you could add a sentence to recommend to use data from streamgages as close as possible to the monitored population; or, the other way round, you could address this issue as a potential limitation of the predictive capability of this approach, vs satellite derived measures which are instead by definition insensitive to this potential issue.

7. PLOS authors have the option to publish the peer review history of their article (what does this mean?). If published, this will include your full peer review and any attached files.

Reviewer #3: **Yes: **Rudy Brogi

Reviewer #4: No

---

## [Author Response · Author response to Decision Letter 2]

26 Oct 2021

These responses are contained in our 'response to reviewers' letter.

---

## [Decision Letter · Decision Letter 3]

1 Dec 2021

PONE-D-20-40787R3Testing the potential of streamflow data to predict spring migration of an ungulatePLOS ONE

Dear Dr. Alexander,

Thank you for submitting your manuscript to PLOS ONE. After careful consideration, we feel that it has merit but does not fully meet PLOS ONE’s publication criteria as it currently stands. Therefore, we invite you to submit a revised version of the manuscript that addresses the points raised during the review process.

We look forward to receiving your revised manuscript.

Kind regards,

Stefano Grignolio, Ph.D

Academic Editor

PLOS ONE

Journal Requirements:

Additional Editor Comments (if provided):

I appreciated the great efforts made by Authors to increase the quality of this manuscript. I fully agree with the positive reviewers' comments. Nevertheless, both reviewers, mainly #3, proposed some comments to improve the manuscript quality. I ask to reply to these comments, paying particular attention to the concern about the statistical analysis and the suggestion about the exclusion of results on MI. Moreover, I will appreciate to read a discussion section less focused on USA situation, in order to increase the interest of this manuscript for a broader audience.

Reviewers' comments:

Reviewer's Responses to Questions

**Comments to the Author**

1. If the authors have adequately addressed your comments raised in a previous round of review and you feel that this manuscript is now acceptable for publication, you may indicate that here to bypass the “Comments to the Author” section, enter your conflict of interest statement in the “Confidential to Editor” section, and submit your "Accept" recommendation.

Reviewer #3: (No Response)

Reviewer #4: All comments have been addressed

2. Is the manuscript technically sound, and do the data support the conclusions?

Reviewer #3: Yes

Reviewer #4: Yes

3. Has the statistical analysis been performed appropriately and rigorously? 

Reviewer #3: N/A

Reviewer #4: Yes

4. Have the authors made all data underlying the findings in their manuscript fully available?

Reviewer #3: Yes

Reviewer #4: Yes

5. Is the manuscript presented in an intelligible fashion and written in standard English?

Reviewer #3: Yes

Reviewer #4: Yes

6. Review Comments to the Author

Reviewer #3: Testing the potential of streamflow data to predict spring migration of an ungulate

PONE-D-20-40787R1

The authors made a great job and addressed most issues I presented in the previous review step, substantially enhancing the clarity of the manuscript. I believe that only one point remains problematic, and it may be addressed to further improve the manuscript quality before publication. Moreover, I have a facultative suggestion to highlight a further (minor) result of the manuscript. Below you can find my original comments of the previous reviewing step (only those not fully solved), authors’ response, and my new comment.

- ORIGINAL COMMENT

Introduction and Discussion chapters miss information regarding cues of ungulate migrations. The reported past and future predictions on mule deer migration onset is justified only if environmental cues (temperature, snowmelt, vegetation growth…all variables that authors cleverly summarized within the streamflow) are the only (or main) cue triggering migration onset. Nevertheless, phenology of northern latitudes ungulate species typically (mostly) relies on photoperiodism, which is independent form environmental changes. This is the reason for global change-driven mismatches between migration and optimal foraging conditions provided within the green wave (e.g., Stenseth and Mysterud 2002; Mysterud 2013) and the consequential loss of fitness in migrating ungulates (mentioned by the authors in L46). Notwithstanding, there are evidence of high environmental plasticity of the migration timing, for example in caribou which mostly trigger migration with snowmelt (Laforge et al. 2021). The results of this manuscript potentially provide a similar evidence, but this issue should be introduced and discussed within its broader chronobiological context.

AUTHORS

The reviewer is correct that photoperiodism plays a role in migration of many

species. However, if photoperiodism were a primary driver of ungulate migrations in

high latitude or high-altitude areas, there would not be such plasticity in migration

timing observed in many ungulate species because the timing of changes in daylight

length at any latitude is essentially constant over geologically short periods. There is

abundant evidence that many migratory ungulates in northern latitudes exploit

resource waves (the so-called ‘green wave’) along latitudinal and elevational

gradients to maximize nutritional uptake in the spring. Merkle et al. (2016), for

example, showed that 7 of 10 species they examined were surfers of the green wave.

• In mule deer of Wyoming, there is additional evidence that, although they could

complete their migration in a matter of days, these animals spend weeks at stopover

feeding grounds along their routes, tracking the highest nutritional content of the

resource (Sawyer and Kauffman, 2011). There is also evidence that, even during

droughts when spring is shorter and the green wave moves faster, mule deer can

track the most nutritionally rich edge of the resource wave (Aikens et al., 2020).

• We recognize that the reviewer’s comment is likely a reflection of our poor framing

of the problem in the Introduction, and a lack of detail in the discussion. We have

put the following language into the Introduction and Discussion in the hopes that it

improves the set-up of the problem and enriches the discussion. New language is

highlighted in yellow, and location is identified by section and paragraph:

o [INTRODUCTION, first paragraph]: “Exploitation of resource waves has been

widely documented as a crucial component in the life history of migratory

herbivores [1]. In northern latitudes, so-called ‘green-wave’ migrations are

caused by spatially-progressing greening of the landscape along elevation and

latitude gradients associated with plant phenological development in the

transition from spring to summer [2-5]}. The green wave hypothesis is a subset

of the broader forage-maturation hypothesis which states that ungulates trade

off forage quantity for forage quality to maximize energy intake [6]. In

mountainous and high latitude regions of North America, the onset of the green

wave coincides with rising daily mean temperatures causing snowmelt and subsequent progressive exposure of soil and vegetation along elevation

gradients.”

o [INTRODUCTION, second paragraph]: “The satellite-derived normalized-

vegetation difference index (NDVI) and associated derivatives, for example have

been used to model spatial gradients in plant crude protein, making inference

on benefits of surfing strategies such as trailing, surfing, or jumping the green

wave [3,9-11]. Bischof et al (2012) used satellite-derived NDVI to observe that

male and female red deer (Cervus elaphus) moved rapidly from summer to

winter grounds, jumping the green wave. Similarly, LaForge et al. (2021) [12]

used time-series of the instantaneous rate of green up (IRG), an NDVI-derived

metric, and observed that female caribou (Rangifer trandus) jumped the

greenwave by 1 week, and maximum greenup in summer feeding grounds

occurred after calving. Alternatively, Merkle et al. (2016) [2] examined surfing

behaviors of a host of ungulates in mountain regions of the western United

States and found that, despite some variation, a majority of species migration

paths tracked high IRG areas (i.e. surfed the green wave).”

o [INTRODUCTION, third paragraph]: “Satellite-derived data such as NDVI, IRG,

and snowcover, however, have a limited period-of-record and can have high-

latency at the daily time step due atmospheric interference (i.e. MODIS). While

typically less limited in period-of-record, temperature-derived measures…”

o [DISCUSSION, NEW third paragraph]: ““There are inherent assumptions in our

models which add complexity to interpretation of our results. The modeled

prediction of a median shift in migration initiation of 8 days earlier than the

mid-20th century for groups of female mule deer is just beyond the calculated

error of the best model (RMSE = 7.4 days). That model relied only on the timing

of snowmelt initiation, and thus was strongly influenced by the underlying, but

significant, temporal trends of that data. Modeling temporal trends in the

timing of female mule deer migration over the past 85 years required

assumptions that these herds have been responding to the same environmental

cues over that period, that no major shifts in migratory behaviors have occurred

(i.e. switch between resident and migratory), and that locations and distances

between each herd's summer and winter feeding grounds has not changed.

Given that land uses, land and herd management, and vegetation across this

landscape has undoubtedly changed over that time period, this assumption is

potentially naive. Research by Sawyer et al. (2019) [37] has shown that some

ungulates, particularly mule deer in the mountainous western United states,

have little plasticity in migratory routes, indicating some of our assumptions are

reasonable for these species, but may not hold for species with higher migratory

route plasticity. Recent modeling by Barnhardt et al. (2020) [38] has indicated

that snowmelt timing and rate are modulated by plant-available water and

available energy, which adds complexity to the assumption that the response of

spring plant phenology to snowmelt has remained constant over the modeled historical period. Nonetheless, the extensive and continuous periods of record at

some streamgages invites potential for investigating temporal changes in

migration initiation that may assist in understanding population dynamics of

migrating ungulates which exploit the green wave.”

NEW COMMENT

Well done! I consider this point fully addressed from my point of view. I only have a facultative proposal to highlight another biological meaning of your results. I would suggest adding a consideration in discussion, pointing out that your data also provided further evidence of a northern ungulate regulating its phenology with environmental cues. This issue is indeed gaining growing attention by researchers in very recent times, for instance:

Laforge, M. P., Bonar, M., & Vander Wal, E. (2021). Tracking snowmelt to jump the green wave: phenological drivers of migration in a northern ungulate. Ecology, 102(3), e03268.

Brogi, R., Merli, E., Grignolio, S., Chirichella, R., Bottero, E., & Apollonio, M. (2021). It is time to mate: population-level plasticity of wild boar reproductive timing and synchrony in a changing environment. Current Zoology.

- ORIGINAL COMMENT

The onset of spring migration by individuals (MI) is biased by the variability in the number of individuals monitored for each year and herd. With the described method, larger samples will inevitably have a higher likelihood to pick smaller minimum values (in this case, earlier dates). MIji is in other words not comparable among years and herds (and indeed seems anticipated in SAR, the herd with more monitored individuals). As individual and herd migration dates are inconsistent (i.e., when comparing SAR and NAR) and predictions have been rightly based only on herd migration (MH) I would suggest removing any mention of MI from the manuscript.

[AUTHOR RESPONSE]:

• The reviewer is correct in their assertion that there is potential for bias in migration

date between individuals in each herd based on sample size.

• However, we would assert that the smaller sample size is more likely to affect the

precision of the estimate, rather than the accuracy.

• If the estimates of the earliest migration day were affected by sample size, then that

would also be true of estimates within each herd. So, to help alleviate the concerns

of bias by the reviewer, we bootstrapped estimates of the mean-minimum (earliest)

migration day for the southern herd by sampling migration initiation days (with

replacement, 100 iterations) using the same sample size of the northern herd for

each year. The results of the experiment are shown in the plot below.

• Two patterns emerge in the plot above:

(1) Within each herd, sample size appears to have only a weak effect on the

estimate of the earliest migration day (i.e. larger samples within each herd

do not necessarily mean earlier migration dates.). For example, there is a

range of nearly 30 days for samples sizes ~15 in the northern herd, and a

range of nearly 50 days for samples sizes ~20 in the southern herd. For

very small sample sizes (n=1), there is some leverage on the migration

day, but overall, the pattern is weak.

(2) The blue hollow circles corresponding to the sample size of the red dots

show the bootstrapped estimate of the mean-earliest migration day

(southern herd) for the same year and same sample size as the red dots

(northern herd). In 7 out of 8 cases, the bootstrapped means of the

southern herd (hollow blue circles) are earlier than the corresponding

means of the same sample size for the northern herd (red dots). The

mean of the bootstrapped earliest day across all years was, in fact, later

than the mean of the observed, but only by 6 days, which is within the

RMSE of the logistic model reported in the text, and thus within the

precision of the method. Likewise, the mean of the maximum earliest bootstrapped estimates (shown as the upper edge of the purple polygon

in the plot) does not overlap with the mean of observed values for the

northern herd, again indicating the observed earlier migration times of

the southern herd are real.

(3) We recognize that the concerns of the reviewer are also likely to arise

with the journal audience. To address concerns of bias due to sample size

we have inserted the following statement into the discussion. Changes to

the text are highlighted in yellow:

a. [DISCUSSION, first paragraph] “The models included in the

analysis were, intentionally, simple, and were constructed mainly to

compare performance of models using streamflow records with

those using more traditional measures of spring to predict past

and future green-wave migration timing of female mule deer.

Given the limited data available data, the models appear to have

also accurately reproduced observed migration behaviors among

groups and individuals within the NAR and SAR herds, with the

best models predicting individuals in the SAR herd have higher

probability of initiating migration earlier in the year. It is possible

that the larger sample size of the SAR herd in most years biased

those data toward earlier migration dates. However, a bootstrap

analysis using the sample size of the northern herd indicated

sample size had little effect on the mean earliest migration

initiation timing in the SAR relative to the NAR.”

NEW COMMENT

I acknowledge authors’ effort to run the bootstrap, but I think that this point is still problematic. The assertion that the smaller sample size is more likely to affect the

precision of the estimate, rather than the accuracy, does not apply here since authors did not calculate the MEAN but the MINIMUM date in which the first individual start migrating. Results of bootstrap are, in my opinion, not fully satisfying. Yes, the average date of SAR seems to remain anticipated in respect to NAR watching bootstrap simulations for over 15 collared animals, but NAR population had typically fewer monitored individuals (min 1, max 16, most years below 11, Tab. 1). Thus, the most informative part of the plot is that below 11 collared individuals, which seems to show a substantial overlap between NAR and SAR. Moreover, a general pattern of decreasing date of migration for increasing number of collared individuals (either real or bootstrapped) seems to emerge for SAR in the provided plot.

Finally, the differences between SAR and NAR migration dates are substantial only for MI and almost negligible for MH (Fig 5 A vs B). How do authors explain this inconsistency among the two measures? Since predictions have been rightly based only on herd migration (MH), my suggestion remains to remove any mention of MI from the manuscript.

Reviewer #4: I am happy with the response of the authors to my comments. Reading the ms again, I only have three very minor comments:

L31 ‘Since ‘host’ is typically used in the field of disease ecology to denote the host for pathogens, I would attain to a more intuitive expression, e.g. ‘…found that several/many ungulate…”

L193: 150 or 151?

L416: thaytrecord? I guess is a typo...

7. PLOS authors have the option to publish the peer review history of their article (what does this mean?). If published, this will include your full peer review and any attached files.

Reviewer #3: **Yes: **Rudy Brogi

Reviewer #4: No

---

## [Author Response · Author response to Decision Letter 3]

14 Dec 2021

Please see the 'response to reviewers' document for our responses to reviewer suggestions.

---

## [Editor Report · Decision Letter 4]

19 Dec 2021

Testing the potential of streamflow data to predict spring migration of an ungulate

PONE-D-20-40787R4

Dear Dr. Alexander,

We’re pleased to inform you that your manuscript has been judged scientifically suitable for publication and will be formally accepted for publication once it meets all outstanding technical requirements.

Kind regards,

Stefano Grignolio, Ph.D

Academic Editor

PLOS ONE

---

## [Editor Report · Acceptance letter]

21 Dec 2021

PONE-D-20-40787R4 

Testing the potential of streamflow data to predict spring migration of an ungulate herds 

Dear Dr. Alexander:

I'm pleased to inform you that your manuscript has been deemed suitable for publication in PLOS ONE. Congratulations! Your manuscript is now with our production department. 

Kind regards, 

on behalf of

Dr. Stefano Grignolio 

Academic Editor

PLOS ONE